# Public Health Messages Associated with Low UV Index Values Need Reconsideration

**DOI:** 10.3390/ijerph16122067

**Published:** 2019-06-12

**Authors:** Maria Lehmann, Annette B. Pfahlberg, Henner Sandmann, Wolfgang Uter, Olaf Gefeller

**Affiliations:** 1Department for Medical Informatics, Biometry and Epidemiology, Friedrich-Alexander-Universität Erlangen-Nürnberg (FAU), Waldstr. 6, 91054 Erlangen, Germany; maria.lehmann@fau.de (M.L.); annette.pfahlberg@fau.de (A.B.P.); wolfgang.uter@fau.de (W.U.); 2uv-tech consulting, Tönniesstr. 9, 24106 Kiel, Germany; sandmann@uv-tech.de

**Keywords:** ultraviolet rays, health promotion, radiation monitoring, sun protection, evidence-based public health

## Abstract

Overexposure to ultraviolet (UV) radiation is the main modifiable risk factor for skin cancer. The Global Solar Ultraviolet Index (UVI) was introduced as a tool to visualize the intensity of UV radiation on a certain day, which should enable and encourage people to take appropriate protective measures. The ‘low’ exposure category of the UVI, defined by a rounded UVI value of 0, 1 or 2, was linked to the health message ‘No protection required’ by the World Health Organization and partner organizations. However, published evidence corroborating this advice is not available. To evaluate the erythemal risk of low UVI days, we analyzed 14,431 daily time series of ambient erythemal irradiance data measured at nine stations of the German solar UV monitoring network during the years 2007–2016. We analyzed the proportion of days in the sample for which ambient erythemal doses calculated for various time intervals exceed average minimal erythemal doses (MEDs) of the Fitzpatrick skin phototypes I–VI to assess the potential for erythema arising from sun exposure on days with low UVI values. Additionally, we calculated for each day the minimum exposure duration needed to receive one MED. Our results indicate that on days with a UVI value of 0, risk of erythema is indeed negligible. Conversely, the abovementioned health message appears misleading when melano-compromised individuals (skin type I and II) spend more than 1.5 hours outdoors on days with a UVI value of 2. Under rare circumstances of prolonged exposure, MEDs of the two most sensitive skin types can also be exceeded even on days with a UVI value of 1. Hence, current WHO guidance for sun protection on days with low UVI values needs reconsideration.

## 1. Introduction

Ultraviolet (UV) radiation, approaching Earth every day in the form of sunlight, has officially been classified as carcinogenic to humans by the International Agency for Research on Cancer [1]. Overexposure to UV radiation is responsible for a substantial proportion of melanoma and nonmelanoma skin cancers [2], the incidence rates of which have been increasing for decades worldwide [3,4]. This implies that skin cancer is largely preventable using appropriate sun protection. Trying to exploit this preventive potential, two crucial questions have been addressed by a variety of approaches [5,6,7,8,9]: “When is sun protection needed?” and “How can it be effectively implemented?” The former aspect requires people to identify situations calling for different levels of sun protection. This is hampered by humans’ lack of a sensory organ for UV radiation and the positive correlation between temperature and UV radiation being too low [10,11] to allow substituting temperature as a valid proxy for the level of UV radiation in daily life. 

To provide an easily understandable measure of the intensity of solar UV radiation for the public, the World Health Organization (WHO), the World Meteorological Organization (WMO), the United Nations Environment Programme (UNEP) and the International Commission on Non-Ionizing Radiation Protection (ICNIRP) introduced the Global Solar UV Index (UVI) in 1995 [12], based on an earlier proposal developed by a Canadian government agency [13]. This index is a unitless quantity (reported as rounded to the nearest integer), proportional to the daily maximum 30 min moving average of the intensity of erythemally weighted [14,15] solar UV irradiance (E_er_) at surface level [16]. The primary purpose of the UVI at the time of its launch was to create an internationally standardized concept for monitoring of long-term changes in ground-level UV radiation, but its additional public health use was already on the agenda [17]. However, it was not until 2002 that WHO and its partner organizations published a practical guide providing information on how the concept of the UVI could be extended to serve as a public awareness tool [16]. The UVI guidance included a classification of the UVI scale into five exposure categories which were linked to specific health messages (Figure 1). 

These messages are intended to be valid for various exposure durations and skin phototypes but focus on fair-skinned people [16]. The exposure category ‘low’, comprising rounded UVI values from 0 to 2, is related to the simple messages ‘No protection required’ and ‘You can safely stay outside!’ As these messages imply harmlessness of sun exposure on such days, there should be a solid evidence base for this clear-cut threshold to not lull people into a false sense of security. Unfortunately, no explanation on how the classification of the UVI scale into the different exposure categories was derived and how the adequacy of the linked health messages was validated has been given. The aim of our study was thus to evaluate potential erythemal effects of exposure to solar UV radiation on days with low UVI values, especially considering the difference in susceptibility to UV radiation-induced damage between distinct skin phototypes. To this end, we analyzed a large high-quality dataset of measurements of erythemal irradiance on days belonging to the low exposure category to quantify ambient erythemal UV doses attained during these days in detail.

## 2. Materials and Methods 

### 2.1. Data Sources

Our dataset consists of measurements of diurnal courses of ambient E_er_ over the ten-year period from 2007 until 2016. Those were conducted by nine stations of the German solar UV monitoring network, which is managed by the German Federal Office for Radiation Protection and the German Federal Environmental Agency. Further institutions associated with this UV monitoring network are the German Weather Service, the Federal Institute for Occupational Safety and Health, the Department for Medical Climatology at the University of Kiel, the Labor Inspectorate of Lower Saxony, and the Bavarian Environmental Agency.

As can be seen in Figure 2, the measuring stations belonging to the network are located all over Germany, ranging from a geographical latitude of 47.91°N to 54.92°N, a geographical longitude of 7.91°E to 14.11°E, and a height above sea level of 4 to 1205 m [18,19]. The geographic distribution of the stations was chosen in such a way that all major climate regions in Germany were covered: Germany’s North Sea coast (with pure marine air) and Baltic Sea coast, the North German Plain, the Central Uplands, the Ruhr district and the Rhine valley (both of which suffer from anthropogenically polluted air), and the Alpine foreland. In this way, the locations of the measuring stations provide a representative sample of places in Germany where people expose themselves to the sun in their working life or in their leisure time.

At each station, measurements of solar UV spectra (leading to E_er_ after weighing with the erythema reference action spectrum [14,15]) were conducted every 6 min each day between sunrise and sunset. The daily UVI value was then calculated by taking the maximum 30 min moving average of E_er_, multiplying it by a constant equal to 40 m²/W, and rounding this to the nearest integer using the “round half up” approach. Double monochromator spectroradiometers (DTM300 or DM150, Bentham, Reading, UK) were used for undertaking the UV measurements. The concept for quality assurance of these measurements includes both regular wavelength calibrations of the spectroradiometers by means of mercury lamps and calibration of the systems’ sensitivity with calibration standards (150 W or 200 W halogene lamps), traceable to secondary standards of the German National Metrology Institute (Physikalisch Technische Bundesanstalt (PTB), Brunswick, Germany). Calibrations are carried out with an interactive calibration software designed specifically for the UV monitoring network. Further details on the stations, the measuring instruments, and criteria for technical quality that were applied to the data can be found elsewhere [20].

As similar UVI conditions can occur concurrently at multiple stations of the network, several calendar days appear more than once in our dataset. Therefore, from here, the term ‘day’ in this paper does not refer to a single specific calendar day, but rather to our unit of observation, which is one diurnal course of E_er_ during this calendar day at one specific station of the network.

### 2.2. Statistical Analysis

To enhance comparability between data from different stations and to compensate for annual variation of solar noon at each station, we transformed the time base of the data from Coordinated World Time to Local Solar Time (LST), where solar noon always occurs at 12:00. To analyze potential UV hazards on days with low UVI values, erythemal irradiance data were linearly interpolated and integrated over certain time intervals to calculate erythemal doses (H_er_) received during that period. The considered intervals were defined as (equivalently to [20]) 11:45–12:15, 11:30–12:30, 11:00–13:00, 10:30–13:30, and 10:00–14:00, corresponding to 0.5, 1, 2, 3, and 4 h centered around solar noon, respectively. Ambient erythemal doses were also calculated for the intervals 08:00–10:00, 14:00–16:00, 07:30–10:30, and 13:30–16:30, i.e., 2 h and 3 h intervals, each centered 3 h before and after noon, respectively. Additionally, the erythemal dose for the total day (sunrise to sunset) was computed.

Descriptive information on the distribution of erythemal doses is reported as median accompanied by the 10th (p10) and 90th (p90) percentile, taking account for skewness of the data. To assess the potential hazard from these doses we report the percentages of days in our sample for which average minimal erythemal doses (MEDs) of Fitzpatrick skin types I through VI [21], as shown in Table 1, are exceeded in the given time intervals. One MED is the amount of (solar) UV exposure, which produces minimal perceptible reddening of the skin (solar erythema) 24 h after exposure. Hence, one MED can be considered a short-time maximum dose that should not be exceeded to prevent detrimental effects of UV radiation on the human body [22].

In a second step, we calculated for each day the minimum exposure duration to receive one MED, t_MED,min_ (again, for all six skin types). We obtained t_MED,min_ by computing all time intervals to exceed one skin type specific MED using an integration procedure with variable lower and upper bounds, and then taking the interval with the minimal length for each day. The distributions of t_MED,min_ are presented as boxplots. Trends in t_MED, min_ with regard to skin type were assessed using the nonparametric Jonckheere–Terpstra trend test. *p*-values less than 0.05 (two-sided) were considered significant. Data analysis was performed using the statistical software package R (Version 3.5.1, R Foundation for Statistical Computing, Vienna, Austria) [23].

## 3. Results

### 3.1. Dataset Description

Our final dataset consisted of erythemal irradiance data of 4961 days with a rounded UVI value of 0 (UVI 0 days), 6117 days with a rounded UVI value of 1 (UVI 1 days), and 3353 days with a rounded UVI value of 2 (UVI 2 days). 

The distributions of UVI 0, UVI 1, and UVI 2 days with regard to the year of their occurrence are given in Table 2. The maximum number of UVI 0 days originates from 2009 (N = 570; 11.5%), the minimum number of such days occurred in 2014 (N = 447; 9.0%). The frequency of UVI 1 days in our sample is highest for the year 2010 (N = 694; 11.3 %) and lowest for the year 2014 (N = 532; 8.7 %). The distribution of UVI 2 days has its maximum in 2008 (N = 427; 12.7%) and its minimum in 2009 (N = 258; 7.7%).

The monthly distributions of UVI 0, UVI 1, and UVI 2 days are given in Table 3. The unimodal distribution of UVI 0 days with its maximum in December (N = 1949; 39.3%) is due to the fact that the winter solstice with greatest solar zenith angle (SZA) is around December 21 on the Northern hemisphere. Monthly distributions of UVI 1 and UVI 2 days, respectively, are bimodal with the two maxima for UVI 1 days occurring in February (N = 1526; 24.9%) and November (N = 1281; 20.9%) and for UVI 2 days in March (N = 1061; 31.6%) and October (N = 913, 27.2%). The frequency of UVI 0 and UVI 1 days during spring and summer months (with seasons defined astronomically [26], i.e., March 20/21 until September 22/23) is very low (0.4% and 7.7% respectively); only UVI 2 days are represented more often (37.0%) in our sample during this period.

### 3.2. Erythemal Doses and Proportion of Days Exceeding MEDs for Certain Time Intervals

Table 4 shows the distribution of ambient erythemal doses received during fixed time intervals and the proportion of days for which one MED of skin types I to VI is exceeded in these intervals. 

Expectedly, exposure loads received either before or after noon yield a smaller erythemal dose than an interval of the same duration around noon for all considered UVI values. Further, due to the shift of our data to LST for normalization, intervals of equal length before and after noon lead to almost exactly equal doses. 

The shortest interval considered, 30 min around noon, does not yield doses exceeding MEDs of any skin type on any day. The 1 h interval around noon does so only for the MED of skin type I on 7.75% of UVI 2 days.

On UVI 0 days, the MEDs of skin types III–VI are never exceeded, and for skin types I and II only for the total day interval and in only 1.23% and 0.04% of eligible days, respectively.

Regarding UVI 1 days, doses received during the 2 and 3 h intervals around noon are already larger than MEDs of the two sensitive skin types II, and especially I, for a considerable proportion (0.49% and 20.08% for skin type II, respectively, and 10.99% and 39.28% for skin type I, respectively) of days. The MED of skin type III is exceeded only for 4.51% of the 4 h intervals around noon and for 32.58% of the total day intervals. The latter interval is also the only one providing doses large enough to exceed the MEDs of skin types IV, V, and VI on 4.72%, 0.07%, and 0.02% of eligible days, respectively.

Concerning UVI 2 days, even 2 h around noon pose a serious erythemal risk for skin type I and II, with nearly 88% and 69%, respectively, of days giving doses exceeding the MED. By contrast, for skin types IV–VI, the same interval does not lead to the excess of one MED for any day in our sample. Erythemal doses from the 4 h interval around noon exceed MEDs of the melano-competent skin types III and IV in more than 80% and about one fourth of days, respectively. The total daily dose is higher than the MEDs of skin types I–IV for more than 90% of all UVI 2 days. Day-long exposure also leads to the MEDs of skin type V and VI being exceeded on 44.77% and about one fifth of UVI 2 days, respectively.

### 3.3. Minimal Exposure Durations to Receive one MED

For UVI 0 days, the daily minimum exposure durations needed to receive one MED (t_MED, min_) was extremely long for skin types I (median 7.2 h) and II (MED exceeded for only N = 2 days, values of t_MED, min_: 9.0 h, 9.3 h). One MED could not be received at all for skin types III–VI. 

The distributions of t_MED, min_ for skin types I–IV on UVI 1 days are shown in Figure 3 and distributions of t_MED, min_ for skin types I–VI on UVI 2 days are shown in Figure 4. t_MED, min_ decreases as the UVI value increases, and the skin type decreases (i.e., becomes more UV sensitive). The distribution of t_MED, min_ is generally strongly right-skewed. 

For UVI 1 days, median t_MED, min_ are quite long, starting from 3.0 h for skin type I and increasing up to 7.7 h for skin type IV. This trend in t_MED, min_ regarding the skin type is significant (*p* < 0.001). One MED of skin type V and VI could only be exceeded on N = 4 and N = 1 UVI 1 days in the sample, respectively (values of t_MED, min_: 8.2, 10.2, 10.8, 10.8, and 12.7 h, respectively). However, for UVI 2 days, median t_MED, min_ is only 1.2 and 1.5 h for skin types I and II, respectively and 2.7, 4.5, 6.5, and 7.35 h for skin types III, IV, V, and VI. Again, this trend could be shown to be significant (*p* < 0.001).

## 4. Discussion

Our analysis of E_er_ measurement data of days with low UVI values from the German solar UV monitoring network demonstrated that MEDs are exceeded on many UVI 2 days after a few hours of outdoor exposure, especially for the fair skin types I and II. Thus, UVI 2 days in mid-latitude regions like Germany carry an erythemal risk for those staying outdoors for longer periods, especially when belonging to the subgroup of melano-compromised skin types. By contrast, MEDs are exceeded for just a negligibly small number of UVI 0 days under extreme exposure conditions and for a considerable number of UVI 1 days after prolonged exposure around noon for the melano-compromised skin types. UVI 0 days therefore do not seem to pose an erythemal risk, while this risk is very limited on UVI 1 days and restricted to specific skin types. The melano-protected skin types V and VI generally show a negligible risk of erythema arising from the low UVI category, as their MEDs could only even be exceeded after nearly day-long exposure on UVI 2 days and, in extremely rare cases, UVI 1 days. Overall, marked differences between skin types in terms of percentages of days for which one MED is exceeded for a given time interval and minimal exposure durations to receive one MED have been illustrated in our analysis, reflecting known differences in susceptibility to UV damages between different skin types.

When the UVI was introduced in 1995, it was already intended to be used as an “integral component of a program to inform the public about UV health risks” [12]. However, a coordinated guidance concerning sun protection measures recommended for specific UVI values was not implemented on the international level at that time. Several local variations of grouping UVI values into categories emerged [27]. In Germany, the Commission on Radiological Protection’s (SSK’s) official recommendation was that only UVI values of 0 and 1 formed the category ‘low’ with sun protection not necessary [28]. This matches our results better than the harmonized recommendations provided by WHO, WMO, UNEP, and ICNIRP in 2002 [16]. These latter criteria, including UVI 2 in the ‘low’ category, were later adopted by SSK to avoid differences in reporting and confusion of the public [29], although the WHO document explicitly states that the UV reporting and sun protection scheme can be varied at the national or local level [16].

In 2012, however, a report from the UVI working group of ICNIRP [17] stated that sun protection might be recommended even at such ‘low’ UV exposure for people who sunburn easily and plan to stay outdoors for prolonged periods. Notwithstanding, the general threshold for recommending sun protection only at UVI levels of 3 and above was reconfirmed, without specific evidence supporting this decision being mentioned. Still, ICNIRP provides guidelines on limits of exposure to UV radiation [24], which claim an exposure limit of 30 J/m² in an 8 h period. The action spectrum used in these guidelines is different from the erythema reference action spectrum of the International Commission on Illumination [14,15] which is used to calculate E_er_. Therefore, this threshold corresponds to 1.0-1.3 SED [25], which would be exceeded in even shorter periods than those needed to receive one MED of the most sensitive skin type I. Although the threshold is aimed primarily at outdoor workers, it is also claimed to be generally valid [24]. ICNIRP itself relaxes this exposure limit by stating that it should be interpreted only as a “desirable goal” for skin exposure (but as a strict limit for ocular exposure) and that it is tailored to the melano-compromised skin types I and II [24]. This limit does not match the UVI health messages and is not mentioned in the official UVI documents. 

A report from the UVI 2015 Workshop in Melbourne, Australia [30], also hints towards a basic problem concerning health risks associated with low UVI exposure: The discussion on this topic focused solely on reviewing evidence of harmful biological effects like DNA damage and immunosuppression owing to sub-erythemal UV exposure, thereby assuming that—contrary to the results of our analysis—exposure effects on low UVI days will be limited to the sub-erythemal level. Still, some countries have implemented sun protection messages for the low exposure category which are more cautious than those given in the WHO recommendation. The United States Environmental Protection Agency has recommended covering up and using sunscreen during low UVI conditions if one burns easily since 2004 [31,32]. The Australian Bureau of Meteorology states on its website that sun protection on days with UVI values of 1 or 2 is “generally not needed unless outside for extended periods” [33]. Both phrases essentially link the low UVI exposure category to ‘low risk’ instead of to ‘no risk’ as proclaimed by the WHO and its partner organizations.

Our analysis, an extension of a pilot study from 2017 [34] and an addition to recent evidence from New Zealand [35,36], implies that recommending sun protection on UVI 2 days should be discussed during a process of updating current public health messages connected to different UVI levels. Unprotected longer outdoor stays on UVI 2 days, especially around noon, should no longer be labeled as harmless. Keeping the current recommendations could be particularly risky for the population in early spring (March) when a lot of UVI 2 days occur and people uncover greater areas of their skin due to rising temperatures although their skin has not yet adapted to higher UV radiation exposure. Future UV guidance should also avoid labeling any level of the UVI as carrying ‘no risk’ as current evidence suggests that there is no threshold dose of UV radiation for the induction of skin cancer and thus no safe limit of exposure [37]. In addition, the adaptation of UVI guidance to different skin types should also be considered. Up to today, a ‘one size fits all’ approach has been used, most probably because of the general notion to keep health messages to the public as simple as possible. Nevertheless, the complex situation in this case may still suggest the use of different health messages for different skin types as the potential risk for erythema ranges from considerable for melano-compromised skin types to negligible for melano-protected skin types on UVI 2 days. When doing so, one has to consider that there is a minority of individuals who cannot be classified according to the Fitzpatrick scale and that the accuracy of self-assessed skin types is limited and should therefore be verified by a dermatologist [38]. Despite these minor impediments, local health authorities could choose those messages suitable for the most sensitive major subgroup of a country or region. A similar solution, though resulting from an analysis primarily focusing on very high instead of low UVI values, has been proposed by other authors already [39]. Both the necessity for local adaptation and the possibility of incorporating skin type and exposure duration in the UVI guidance have already been discussed at the WHO UVI workshop in Melbourne in 2015 [30], but have not yet been implemented. 

The strength of our analysis lies in the evaluation of a large dataset comprising measurement data of 10 consecutive years from 9 measuring stations of a solar UV monitoring network with a well-established system of quality control and a geographical distribution of the stations giving a representative sample of places in Germany where people expose themselves to the sun. In total, 14,431 daily E_er_ time series from days of the ‘low’ UVI category were available. Such an analysis of measurements capturing the variability of diurnal courses of E_er_ under real-world conditions is superior to the mere mathematical derivation of UV doses by modeling the diurnal course of E_er_ through the use of, e.g., Gaussian-like or even constant functions. 

Still, our study suffers from some limitations. For our analysis, we implicitly assumed validity of the Bunsen–Roscoe reciprocity law [40]. That is to say, we implied that the erythemal impact of UV radiation in human skin is directly proportional to the total energy dose but does not depend on the exposure duration needed to apply this dose. Research on this topic is comparatively scarce [41], but in a review from 2003 [42], reciprocity regarding erythema was shown to hold in most studies. Still, this law cannot necessarily be adapted to exposure scenarios incorporating breaks. Repair mechanisms could lead to doses resulting from interrupted exposure being subadditive, whereby sums of doses from our analysis, e.g., for intervals in the morning and the afternoon, could then only be seen as an upper threshold for estimating the actual effect in human skin. 

Our dataset consists of ambient erythemal irradiance data which are measured on horizontal detectors. Due to this, the measurements potentially just weakly approximate individual exposure as most human skin surfaces are not oriented horizontally. On the one hand, surfaces facing the sun (almost) vertically can receive significantly higher irradiances (up to 40%) during periods without cloud obstruction and with high SZA (i.e., with the sun low in the sky) [43]. The majority of days in our sample (more than 90% of UVI 0 and 1 days and more than 60% of UVI 2 days) originate from the autumn and winter period and are therefore likely to represent this scenario. On the other hand, cloudy conditions can reduce UV on tilted surfaces by up to 50% in comparison to horizontal-incidence UV [43]. Cloudy conditions are likely to have been prevalent on many days in our sample during the late spring and summer period, because only a thick cloud cover sufficiently attenuates UV radiation which would otherwise be much higher during this time of the year. Moreover, the ratio between personal and ambient exposure, frequently called exposure ratio (ER), is highly dependent on individual behavior. This covers aspects like use of shade [44], intermittent indoor activities, and body posture [45]. Values of ER for outdoor workers reported in a review from 2011 ranged from 8–66% (arms and wrists), 11–85% (vertex) and 11–70% (shoulders), respectively [46]. Values for ER are, of course, even lower for indoor workers who spend less time outside in the sun than outdoor workers. A recent study from New Zealand found values of less than 2% for ER of indoor workers [47]. Still, the official UVI guidelines for the low exposure category should also be valid for a ‘worst case’ scenario of staying outside for prolonged periods with no sun protection applied. This scenario is what we apply in our analysis. 

Textile protection of the skin is also not considered in our study. Adequate clothing can protect human skin from deleterious UV exposure, the amount of protection depending on the coverage level of the body surface area, and the quality of UV protection offered by the fabric [48,49]. Most of the days in our sample were in autumn and winter when people—due to low ambient temperatures—tend to cover most of their body by clothes offering a high level of UV protection. Still, the body parts that mostly remain uncovered are the hands and the face, which are both common localizations of skin malignancies [50] and are also oriented vertically while standing. This can, as discussed above, lead to even higher irradiances and doses because of the high SZA in this period of the year. 

Finally, our analysis focuses on negative health-related aspects of UV exposure on days with low UVI values, namely, the induction of erythema. In the course of potentially adapting UVI guidance to our results, positive health-related aspects of UV exposure like production of vitamin D [2,51,52,53] should also be considered.

## 5. Conclusions

Current WHO guidance for sun protection on days with ‘low’ UVI values needs reconsideration. Our analysis revealed that UV exposure for prolonged exposure durations on UVI 2 days and, under certain rare circumstances, even on UVI 1 days, reaches erythemal levels, and thus, sun protection is required to avoid deleterious effects. This particularly relates to sensitive skin types, which might imply the need for skin type specific public health messages relating to the UVI.

## Figures and Tables

**Figure 1 ijerph-16-02067-f001:**
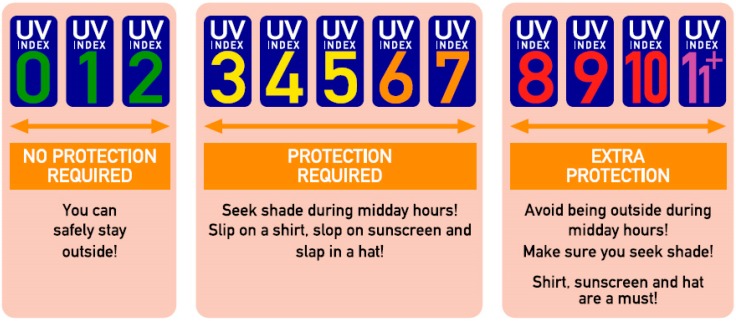
UV Index (UVI) guidance according to the WHO. Figure modified from [16]. The modification consists of the explicit graphical representation of the value zero in the ‘low exposure’ category, which is not part of the original WHO figure but is mentioned to belong to this category in the text [16].

**Figure 2 ijerph-16-02067-f002:**
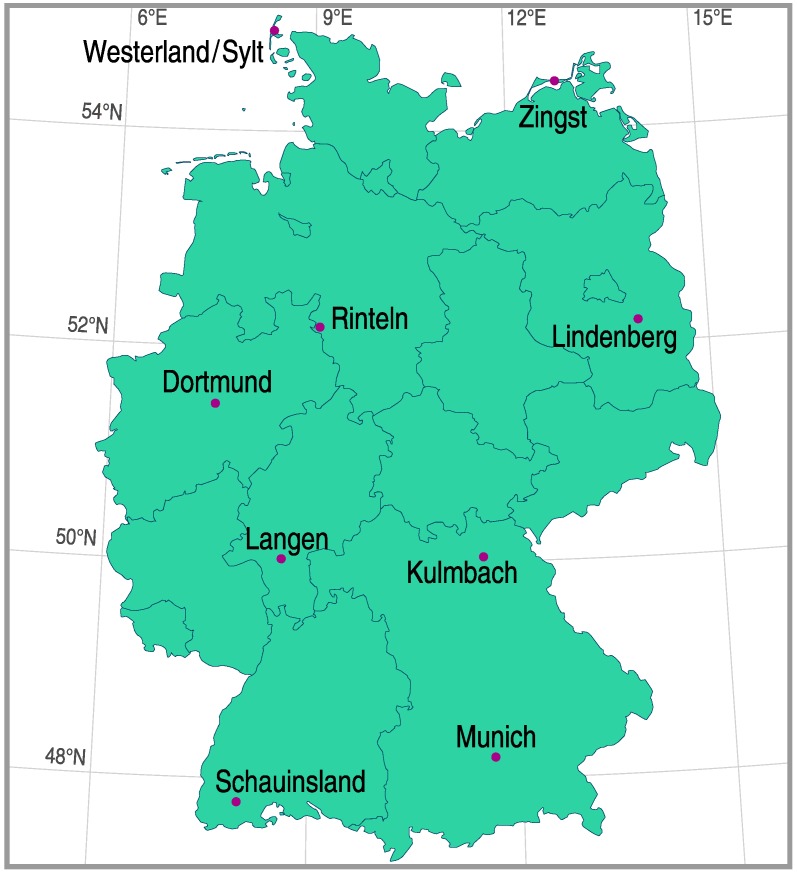
Map of Germany showing the geographical position of the nine stations of the German solar UV monitoring network which provided erythemal irradiance data.

**Figure 3 ijerph-16-02067-f003:**
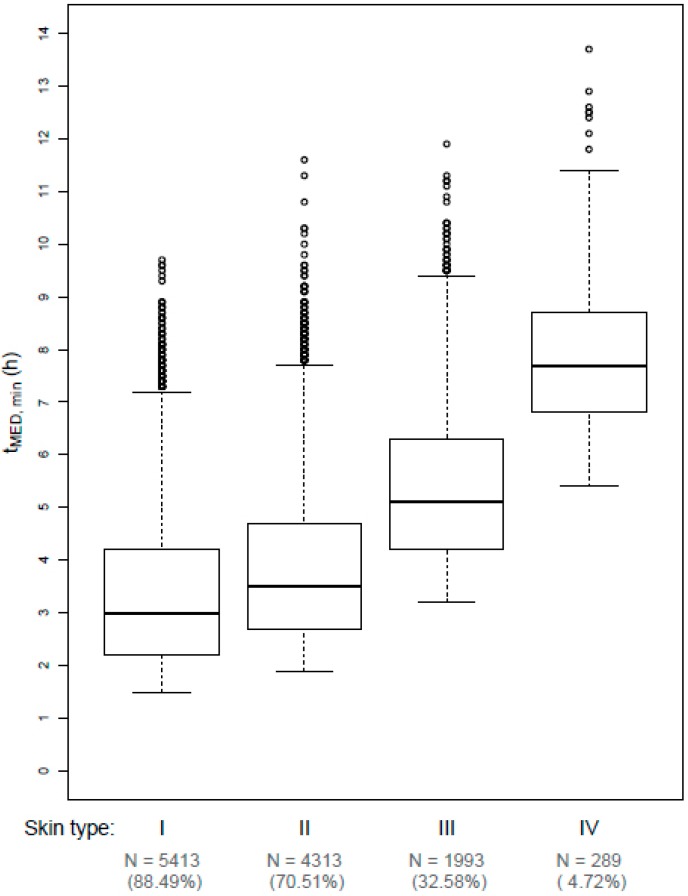
Boxplots of daily minimum exposure durations (in hours) needed to receive one MED (t_MED, min_) for skin types I–IV for days with rounded UVI values of 1. Bold line: median, lower/upper edge of box: 25%/75% quantile (first/third quartile), whiskers: range of data points which are no more than 1.5 times the interquartile range out of the box, dots: outliers. N (%) denotes the absolute (relative) number of observations in the subsample, i.e., the absolute (relative) number of UVI 1 days for which one MED could actually be received. The trend in t_MED,min_ with respect to skin type can be considered significant (*p* < 0.001).

**Figure 4 ijerph-16-02067-f004:**
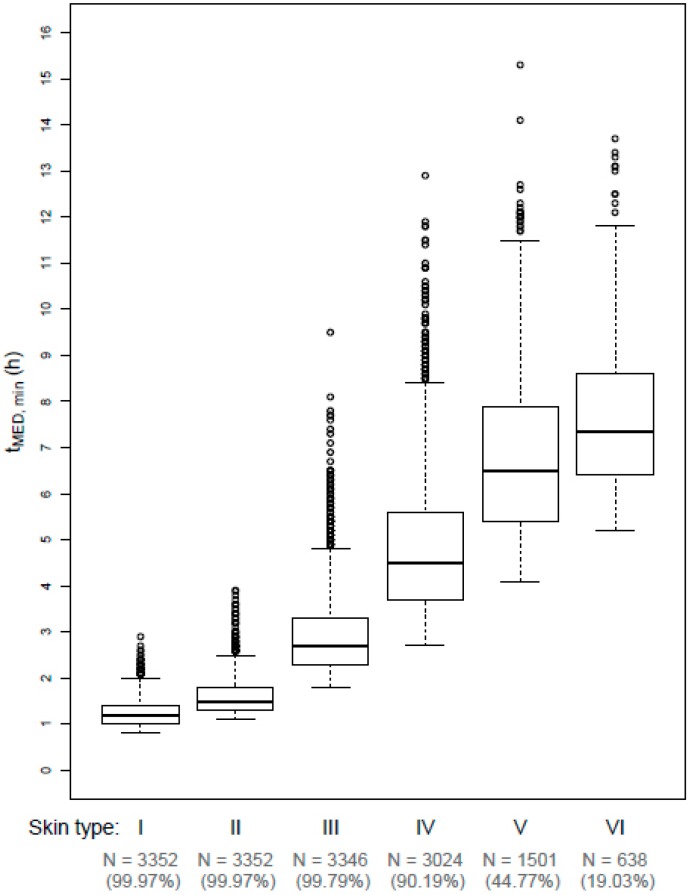
Boxplots of daily minimum exposure durations (in hours) needed to receive one MED (t_MED, min_) for skin types I–VI for days with rounded UVI values of 2. Bold line: median, lower/upper edge of box: 25%/75% quantile (first/third quartile), whiskers: range of data points which are no more than 1.5 times the interquartile range out of the box, dots: outliers. N (%) denotes the absolute (relative) number of observations in the subsample, i.e., the absolute (relative) number of UVI 2 days for which one MED could actually be received. The trend in t_MED,min_ with respect to skin type can be considered significant (*p* < 0.001).

**Table 1 ijerph-16-02067-t001:** Column 1: Fitzpatrick skin types [21], column 2: specific skin response to sun exposure [24], column 3: further classification of the skin types [25], column 4: average minimal erythemal doses (MEDs) corresponding to the skin types according to the International Commission on Non-Ionizing Radiation Protection (ICNIRP) [25]. MEDs given in terms of standard erythema doses (1 SED = 1 standard erythema dose = 100 J/m² weighted with the International Commission on Illumination (CIE) erythema reference action spectrum [14]).

Skin Type	Skin Response to Sun Exposure	Classes of Individuals	MED (in SED)
I	Burns easily and severely (painful burn); tans little or none and peels	Melano-compromised	2.0
II	Usually burns easily and severely (painful burn); tans minimally or lightly, also peels		2.5
III	Burns moderately and tans	Melano-competent	4.0
IV	Burns minimally, tans easily		6.0
V	Rarely burns, tans easily and substantially	Melano-protected	8.5
VI	Never buns and tans profusely		10.0

**Table 2 ijerph-16-02067-t002:** Absolute (N) and relative (%) frequency distributions of UVI 0, 1, and 2 days, according to year of occurrence.

Year	UVI 0		UVI 1		UVI 2	
	N	%	N	%	N	%
2007	526	10.6	616	10.1	343	10.2
2008	509	10.3	585	9.6	427	12.7
2009	570	11.5	534	8.7	258	7.7
2010	533	10.7	694	11.3	361	10.8
2011	482	9.7	618	10.1	307	9.2
2012	486	9.8	650	10.6	332	9.9
2013	488	9.8	632	10.3	322	9.6
2014	447	9.0	532	8.7	324	9.7
2015	453	9.1	604	9.9	340	10.1
2016	467	9.4	652	10.7	339	10.1
Total	4961	100.0	6117	100.0	3353	100.0

**Table 3 ijerph-16-02067-t003:** Absolute (N) and relative (%) frequency distributions of UVI 0, 1, and 2 days, according to month of occurrence.

Month	UVI 0		UVI 1		UVI 2	
	N	%	N	%	N	%
Jan	1515	30.5	742	12.1	1	0.0
Feb	396	8.0	1526	24.9	249	7.4
Mar	54	1.1	674	11.0	1061	31.6
Apr	3	0.1	92	1.5	300	8.9
May	1	0.0	69	1.1	136	4.1
Jun	0	0.0	29	0.5	78	2.3
Jul	6	0.1	28	0.5	71	2.1
Aug	0	0.0	44	0.7	105	3.1
Sep	10	0.2	145	2.4	377	11.2
Oct	88	1.8	1047	17.1	913	27.2
Nov	939	18.9	1281	20.9	62	1.8
Dec	1949	39.3	440	7.2	0	0.0
Total	4961	100.0	6117	100.0	3353	100.0

**Table 4 ijerph-16-02067-t004:** Ambient erythemal UV doses (H_er_) calculated for different time intervals on days with a UVI value of 0, 1, and 2 and corresponding proportion of days with doses from those intervals exceeding one MED for the Fitzpatrick [21] skin types I to VI. (1 SED = 1 standard erythema dose = 100 J/m² weighted with the CIE erythema reference action spectrum [14]).

Time Interval (Local Solar Time, Duration)	H_er_ (in SED); Median (p10, p90 Percentile)	Proportion of Days Exceeding one MED for Skin Type (in %)					
		I	II	III	IV	V	**VI**
Before noon							
8:00–10:00, 2 h							
UVI 0	0.12 (0.05, 0.23)	0	0	0	0	0	0
UVI 1	0.42 (0.20, 0.86)	0.07	0	0	0	0	0
UVI 2	1.24 (0.69, 1.92)	7.87	2.09	0	0	0	0
7:30–10:30, 3 h							
UVI 0	0.19 (0.09, 0.36)	0	0	0	0	0	0
UVI 1	0.67 (0.34, 1.31)	1.03	0.15	0	0	0	0
UVI 2	1.90 (1.11, 2.85)	44.32	19.92	0.89	0	0	0
Around noon							
11:45–12:15, 0.5 h							
UVI 0	0.11 (0.05, 0.19)	0	0	0	0	0	0
UVI 1	0.32 (0.20, 0.54)	0	0	0	0	0	0
UVI 2	0.74 (0.43, 1.00)	0	0	0	0	0	0
11:30–12:30, 1 h							
UVI 0	0.23 (0.11, 0.37)	0	0	0	0	0	0
UVI 1	0.64 (0.40, 1.05)	0	0	0	0	0	0
UVI 2	1.46 (0.93, 1.96)	7.75	0	0	0	0	0
11:00–13:00, 2 h							
UVI 0	0.44 (0.21, 0.70)	0	0	0	0	0	0
UVI 1	1.24 (0.80, 2.03)	10.99	0.49	0	0	0	0
UVI 2	2.83 (1.92, 3.80)	87.89	68.65	5.37	0	0	0
10:30–13:30, 3 h							
UVI 0	0.63 (0.31, 0.99)	0	0	0	0	0	0
UVI 1	1.78 (1.15, 2.90)	39.28	20.08	0	0	0	0
UVI 2	4.08 (2.86, 5.45)	98.39	95.38	52.82	2.42	0	0
10:00–14:00, 4 h							
UVI 0	0.79 (0.39, 1.23)	0	0	0	0	0	0
UVI 1	2.24 (1.43, 3.64)	60.72	39.82	4.51	0	0	0
UVI 2	5.14 (3.71, 6.88)	99.79	99.14	84.01	26.39	0	0
After noon							
14:00–16:00, 2 h							
UVI 0	0.12 (0.05, 0.22)	0	0	0	0	0	0
UVI 1	0.42 (0.20, 0.86)	0.08	0	0	0	0	0
UVI 2	1.24 (0.72, 1.92)	7.99	2.42	0	0	0	0
13:30–16:30, 3 h							
UVI 0	0.19 (0.08, 0.36)	0	0	0	0	0	0
UVI 1	0.67 (0.33, 1.32)	1.32	0.25	0	0	0	0
UVI 2	1.92 (1.16, 2.85)	44.68	20.34	0.95	0	0	0
Total day							
Sunrise–Sunset							
UVI 0	1.04 (0.52, 1.65)	1.23	0.04	0	0	0	0
UVI 1	3.21 (1.95, 5.46)	88.49	70.51	32.58	4.72	0.07	0.02
UVI 2	8.17 (6.01, 10.87)	99.97	99.97	99.79	90.19	44.77	19.03

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
