# Peer review of "Public Health Messages Associated with Low UV Index Values Need Reconsideration"

_ijerph, 2019, doi:10.3390/ijerph16122067_

Reviewer 1 Report

The paper 'Public Health Messages Associated with the Low Exposure Category of the UV Index Need Reconsideration' by Lehmann et al. is well organised and has a clear construction. Methodology seems to be adequate to the considered problem and is mostly clearly described. Results are provided in the form of understandable tables and two graphs. The work aims to prove the hypothesis, that the classification of Global Solar Ultraviolet Index (UVI) by the risk it poses, presented by World Health Organization and others, should be reconsidered. By the analysis of real measurements data of high quality standards authors show, that the category 'no protection required' should be reconsidered. They proved, that UVI 2 can provide eryhemal doses, that are larger of 1 MED for phototypes I and II even in relatively short time spend outdoors (up to two hours). Even UVI 1 can pose a risk to the individuals, who spend most of their day outside, for example outdoor-workers. The results are interesting and should be taken into consideration by WHO to reconsider their UVI classification.

General comments:

The paper is well written, with clear construction. The abstract summarize the work adequately, describes the overall work and contains results. The problem raised by the researches is of a great importance to human health. They present a new approach to UVI classification by focusing not on large values of UVI (as most of the researches does), but on low values, which are believed to not requiring any form of photoprotection. However, there are some inconsistency in the paper that should be clarified. One question is, if erythemal doses cumulate after the break in the exposure duration (for example few hours)? If they do, the authors could consider cumulative doses too, for example 1h around the noon plus 3h in the afternoon. Authors claim, that 'the positive correlation between temperature and UV radiation is too low' but they do not give any references nor calculations to support that statement. It would be relatively easy to calculate such correlation (only for days with UVI 0, UVI 1 and UVI 2) and could improve the manuscript. Otherwise, authors should give the references. Furthermore, authors should clarify how the minimum exposure duration to receive one MED was calculated and add more extensive description to the boxplots.

Detailed comments:

Page 2, line 92: although the authors refer to their previous work with more details, they should add some comment about calibration of instruments in the manuscript;

Page 3, line 111: it appears from the results, that also cumulative doses for the whole day (sunrise-sunset) have been calculated. The authors should mention about it in this place;

Page 3, line 121: authors should clarify how tMED,min was calculated. Was it calculated for the maximum value of erythemal UV during the day? It is not clear;

Page 7, line 203: the last word 'respectively' is not needed here;

Page 9, line 281: authors should mention, that there are some individuals, who cannot be cassified by Fitzpatrick scale. Furthermore, the self-classification to Fitzpartick skin phototypes should be verified by the dermatologists (see for example Eilers S, Bach DQ, Gaber R, et al. Accuracy of Self-report in Assessing Fitzpatrick Skin Phototypes I Through VI. JAMA Dermatol. 2013;149(11):1289–1294. doi:10.1001/jamadermatol.2013.6101).

Page 10, line 329: as authors mention positive health-related aspects of UV exposure, they should add a few references.

Figure 2, Figure 3: values on the boxplots are not described. Authors should describe bold line, edges of boxes, whiskers and dots.

Author Response

Point-by-point-reply for reviewer 1 for manuscript “Public Health Messages Associated with Low UV Index Values Need Reconsideration”, by Maria Lehmann, Annette B. Pfahlberg, Henner Sandmann, Wolfgang Uter and Olaf Gefeller.

We thank the reviewer for his thoughtful and constructive comments on our manuscript that have helped to improve the presentation of our study and its findings. We have prepared a revised and expanded version of the manuscript incorporating his suggestions. We address below the specific points raised by the reviewer and explain how they have been dealt with in the revised manuscript. The highlighted version of the revised manuscript accompanying the submission shows the changes in detail. When we refer to specific page and line numbers in our reply the revised version including the highlighted changes is the basis for these text references.

Open Review

English language and style

( ) Extensive editing of English language and style required
( ) Moderate English changes required
(x) English language and style are fine/minor spell check required
( ) I don't feel qualified to judge about the English language and style

Yes

Can be improved

Must be   improved

Not   applicable

Does   the introduction provide sufficient background and include all relevant   references?

(x)

( )

( )

( )

Is the   research design appropriate?

(x)

( )

( )

( )

Are the   methods adequately described?

( )

(x)

( )

( )

Are the   results clearly presented?

(x)

( )

( )

( )

Are the   conclusions supported by the results?

(x)

( )

( )

( )

Comments and Suggestions for Authors

The paper 'Public Health Messages Associated with the Low Exposure Category of the UV Index Need Reconsideration' by Lehmann et al. is well organised and has a clear construction. Methodology seems to be adequate to the considered problem and is mostly clearly described. Results are provided in the form of understandable tables and two graphs. The work aims to prove the hypothesis, that the classification of Global Solar Ultraviolet Index (UVI) by the risk it poses, presented by World Health Organization and others, should be reconsidered. By the analysis of real measurements data of high quality standards authors show, that the category 'no protection required' should be reconsidered. They proved, that UVI 2 can provide eryhemal doses, that are larger of 1 MED for phototypes I and II even in relatively short time spend outdoors (up to two hours). Even UVI 1 can pose a risk to the individuals, who spend most of their day outside, for example outdoor-workers. The results are interesting and should be taken into consideration by WHO to reconsider their UVI classification.

General comments:

The paper is well written, with clear construction. The abstract summarize the work adequately, describes the overall work and contains results. The problem raised by the researches is of a great importance to human health. They present a new approach to UVI classification by focusing not on large values of UVI (as most of the researches does), but on low values, which are believed to not requiring any form of photoprotection. However, there are some inconsistency in the paper that should be clarified.

One question is, if erythemal doses cumulate after the break in the exposure duration (for example few hours)? If they do, the authors could consider cumulative doses too, for example 1h around the noon plus 3h in the afternoon.

We now explain that for our analysis validity of the Bunsen-Roscoe reciprocity law is assumed (meaning the erythemal effect of UV radiation is only dose-dependent, but not dependent on exposure duration), but that this does not guarantee that the sum of doses from 2 (or more) intervals incorporating an exposure break does have the exact same effect as an interval without breaks (see lines 342 ff.). Therefore we would not like to perform such summations.

Authors claim, that 'the positive correlation between temperature and UV radiation is too low' but they do not give any references nor calculations to support that statement. It would be relatively easy to calculate such correlation (only for days with UVI 0, UVI 1 and UVI 2) and could improve the manuscript. Otherwise, authors should give the references.

We added 2 references (line 45) as we do not have temperature data for the stations of the UV monitoring network.

Beck, N.; Balanay, J.A.G.; Johnson, T. Assessment of occupational exposure to heat stress and solar ultraviolet radiation among groundskeepers in an eastern north carolina university setting. J. Occup. Environ. Hyg. 2018, 15, 105-116.

Morabito, M.; Grifoni, D.; Crisci, A.; Fibbi, L.; Orlandini, S.; Gensini, G.F.; Zipoli, G. Might outdoor heat stress be considered a proxy for the unperceivable effect of the ultraviolet-induced risk of erythema in florence? J. Photochem. Photobiol. B-Biol. 2014, 130, 338-348.

The first one explicitly states an overall correlation coefficient of r=0.422 between wet bulb globe temperature and erythemal irradiance which supports our claim of low correlation.

Furthermore, authors should clarify how the minimum exposure duration to receive one MED was calculated and add more extensive description to the boxplots.

Please see below.

Detailed comments:

Page 2, line 59-60: a table or graph with WHO classification of UVI should be added;

We added a figure (Figure 1, line 61).

Page 2, line 92: although the authors refer to their previous work with more details, they should add some comment about calibration of instruments in the manuscript;

We added detailed information on the calibration of instruments in lines 104 ff.

Page 3, line 111: it appears from the results, that also cumulative doses for the whole day (sunrise-sunset) have been calculated. The authors should mention about it in this place;

Added (line 129).

Page 3, line 121: authors should clarify how tMED,min was calculated. Was it calculated for the maximum value of erythemal UV during the day? It is not clear;

We added one sentence stating that we obtained tMED,min by computing all time intervals to exceed one skin type specific MED using an integration procedure with variable lower and upper bounds, and then taking the interval with the minimal length for each day. (line 142).

Page 7, line 203: the last word 'respectively' is not needed here;

Erased.

Page 9, line 281: authors should mention, that there are some individuals, who cannot be classified by Fitzpatrick scale. Furthermore, the self-classification to Fitzpartick skin phototypes should be verified by the dermatologists (see for example Eilers S, Bach DQ, Gaber R, et al. Accuracy of Self-report in Assessing Fitzpatrick Skin Phototypes I Through VI. JAMA Dermatol. 2013;149(11):1289–1294. doi:10.1001/jamadermatol.2013.6101).

We added this in line 325 ff.

Page 10, line 329: as authors mention positive health-related aspects of UV exposure, they should add a few references.

We added 4 references (line 383):

Lucas, R.M.; McMichael, A.J.; Armstrong, B.K.; Smith, W.T. Estimating the global disease burden due to ultraviolet radiation exposure. Int. J. Epidemiol. 2008, 37, 654-667.

Holick, M.F. Sunlight and vitamin D for bone health and prevention of autoimmune diseases, cancers, and cardiovascular disease. Am. J. Clin. Nutr. 2004, 80, 1678S-1688S.

Autier, P.; Boniol, M.; Pizot, C.; Mullie, P. Vitamin D status and ill health: A systematic review. The Lancet Diabetes & Endocrinology 2014, 2, 76-89.

Zeeb, H.; Greinert, R. The role of vitamin D in cancer prevention: Does UV protection conflict with the need to raise low levels of vitamin D? Deutsches Arzteblatt international 2010, 107, 638-643.

Figure 2, Figure 3: values on the boxplots are not described. Authors should describe bold line, edges of boxes, whiskers and dots.

There are many different conventions for boxplots and we agree that the one we used should therefore be specified in the figure caption. We added one sentence for each figure (lines 228 ff. and 245 ff.).

Submission Date

29 April 2019

Date of this review

13 May 2019 13:13:08

Reviewer 2 Report

The authors describe an analysis on the ultraviolet index which looks at UVI days deemed 0, 1 and 2 (low risk) and how much UV is detected during each day of the year and how this fits with the UVI model.  Overall they suggest that more stringent UVI recommendations be made each day to more precisely fit with skin type.

Major Comments:

Please re-title figures 2 and 3 with a more accurate description of the data. 

Did the authors run any statistical analysis for Figures 2 and 3, if so, please describe in the figure caption.  If not, are statistics appropriate?

Minor Comments:

Line 230:  categorization is not a word, please replace.

Line 277:  Can the authors find another phrase than Keep it Simple stupid here and in line 280?  The language seems inappropriate for a scientific journal.

Author Response

Point-by-point-reply for reviewer 2 for manuscript “Public Health Messages Associated with Low UV Index Need Values Reconsideration”, by Maria Lehmann, Annette B. Pfahlberg, Henner Sandmann, Wolfgang Uter and Olaf Gefeller.

We thank the reviewer for his thoughtful and constructive comments on our manuscript that have helped to improve the presentation of our study and its findings. We have prepared a revised and expanded version of the manuscript incorporating his suggestions. We address below the specific points raised by the reviewer and explain how they have been dealt with in the revised manuscript. The highlighted version of the revised manuscript accompanying the submission shows the changes in detail. When we refer to specific page and line numbers in our reply the revised version including the highlighted changes is the basis for these text references.

Open Review

English language and style

( ) Extensive editing of English language and style required
( ) Moderate English changes required
(x) English language and style are fine/minor spell check required
( ) I don't feel qualified to judge about the English language and style

Yes

Can be   improved

Must be   improved

Not   applicable

Does   the introduction provide sufficient background and include all relevant   references?

(x)

( )

( )

( )

Is the   research design appropriate?

(x)

( )

( )

( )

Are the   methods adequately described?

(x)

( )

( )

( )

Are the   results clearly presented?

(x)

( )

( )

( )

Are the   conclusions supported by the results?

(x)

( )

( )

( )

Comments and Suggestions for Authors

The authors describe an analysis on the ultraviolet index which looks at UVI days deemed 0, 1 and 2 (low risk) and how much UV is detected during each day of the year and how this fits with the UVI model.  Overall they suggest that more stringent UVI recommendations be made each day to more precisely fit with skin type.

Major Comments:

Please re-title figures 2 and 3 with a more accurate description of the data. 

The description of these two figures was indeed lacking a clear description of the definition of the properties of the boxplot. We added this information for both figures (line 228 and 245).

Did the authors run any statistical analysis for Figures 2 and 3, if so, please describe in the figure caption.  If not, are statistics appropriate?

We now include a statistical trend test (the nonparametric Jonckheere-Terpstra trend test) in the analysis of tMED,min. The relevant information can be found in lines 236 and 240 in the text of the results section, in lines 233 and 250 in the figure captions and also in line 145 of the methods section.

Minor Comments:

Line 230:  categorization is not a word, please replace.

We rephrased the sentence avoiding ‘categorization’, see line 271.

Line 277:  Can the authors find another phrase than Keep it Simple stupid here and in line 280?  The language seems inappropriate for a scientific journal.

We agree that this phrase is rather colloquial. We therefore erased it and only describe that public health messages should be kept simple without referring to this principle (line 317 ff.)

Submission Date

29 April 2019

Date of this review

07 May 2019 19:57:05

Reviewer 3 Report

The paper offers an interesting discussion about the efficacy of risk information to the public about potentially dangerous exposure to solar UV radiation when UVI values are low. The manuscript is well structured, with a clear introduction, a substantially detailed presentation of results and an accordingly consistent discussion. Conclusions are corroborated by what discussed in the main body of the paper. References are adequate and updated to the recent literature. I would only highlight that, although low UVI values could lead to erythema under some circumstances, it must be said that such conditions are unlikely in real life. Nevertheless, the work poses a scientifically sounding question and finds an answer. For such reasons, I consider it suitable for publication, after some minor methodological discussions and, eventually, corrections.

At first I would ask the authors if they think the title could be shortened in “Public Health Messages Associated with Low UV Index Need Reconsideration”. This is not substantial but the message to the reader would not be different from the original title.

The MED is a laboratory amount determined by dermatologists and photobiologists as the minimal dose that causes sunburn or redness. The authors compare MED with the median of the erythemal dose (L.112): why do they choose the median?

In the caption of Table 1 (L.123-127) I suggest to add the indication of columns (i.e.  column 1 Fitzpatrick skin types [18], column 2 specific skin response to sun exposure …).

In Results (L.131) the authors say that they determine “rounded UVI value”: what does that mean? What kind of approximation is used? The authors should clarify how they consider, as an example, the value of UVI=2.3: is it considered as 2, so that it belongs to “Low” category, or as above 2 so that it is in the higher risk class?

There’s been a long discussion in the past about the erythemal action spectrum. Although erythemal data were provided by radiometric stations, it would be correct to add to the references the following paper: “Know Your Standard: Clarifying the CIE Erythema Action Spectrum” Ann R. Webb  Harry Slaper  Peter Koepke  Alois W. Schmalwieser (2010) https://doi.org/10.1111/j.1751-1097.2010.00871.x

Author Response

Point-by-point-reply for reviewer 3 for manuscript “Public Health Messages Associated with Low UV Index Values Need Reconsideration”, by Maria Lehmann, Annette B. Pfahlberg, Henner Sandmann, Wolfgang Uter and Olaf Gefeller.

We thank the reviewer for his thoughtful and constructive comments on our manuscript that have helped to improve the presentation of our study and its findings. We have prepared a revised and expanded version of the manuscript incorporating his suggestions. We address below the specific points raised by the reviewer and explain how they have been dealt with in the revised manuscript. The highlighted version of the revised manuscript accompanying the submission shows the changes in detail. When we refer to specific page and line numbers in our reply the revised version including the highlighted changes is the basis for these text references.

Open Review

English language and style

( ) Extensive editing of English language and style required
( ) Moderate English changes required
(x) English language and style are fine/minor spell check required
( ) I don't feel qualified to judge about the English language and style

Yes

Can be   improved

Must be   improved

Not   applicable

Does   the introduction provide sufficient background and include all relevant   references?

(x)

( )

( )

( )

Is the   research design appropriate?

(x)

( )

( )

( )

Are the   methods adequately described?

(x)

( )

( )

( )

Are the   results clearly presented?

(x)

( )

( )

( )

Are the   conclusions supported by the results?

(x)

( )

( )

( )

Comments and Suggestions for Authors

The paper offers an interesting discussion about the efficacy of risk information to the public about potentially dangerous exposure to solar UV radiation when UVI values are low. The manuscript is well structured, with a clear introduction, a substantially detailed presentation of results and an accordingly consistent discussion. Conclusions are corroborated by what discussed in the main body of the paper. References are adequate and updated to the recent literature. I would only highlight that, although low UVI values could lead to erythema under some circumstances, it must be said that such conditions are unlikely in real life. Nevertheless, the work poses a scientifically sounding question and finds an answer. For such reasons, I consider it suitable for publication, after some minor methodological discussions and, eventually, corrections.

At first I would ask the authors if they think the title could be shortened in “Public Health Messages Associated with Low UV Index Need Reconsideration”. This is not substantial but the message to the reader would not be different from the original title.

We agree that the title would be handier if it was shortened. We chose a new title close to the reviewer’s proposal, “Public Health Messages Associated with Low UV Index Values Need Reconsideration”.

The MED is a laboratory amount determined by dermatologists and photobiologists as the minimal dose that causes sunburn or redness. The authors compare MED with the median of the erythemal dose (L.112): why do they choose the median?

The reviewer correctly hints towards the fact that the median dose does indeed not have any special meaning for public health and this comparison was therefore only moderately useful. We erased this type of comparison from the results section, but left the information on median and p10 and p90 percentile in table 4 to give an adequate impression of dose distributions. To this end, we also kept the paragraph starting from line 186 dealing with general properties of the doses from the considered intervals without their effect on the specific skin types. This effect is still dealt with in form of reporting the proportion of days for which the skin type specific MED is exceeded for the different intervals.

We also undertook changes in the abstract (line 21) and the methods section (starting from line 130) reflecting these changes.

In the caption of Table 1 (L.123-127) I suggest to add the indication of columns (i.e.  column 1 Fitzpatrick skin types [18], column 2 specific skin response to sun exposure …).

We agree that adding the indication of columns would increase comprehensibility here and therefore added this information in the table caption (lines 149 ff.)

In Results (L.131) the authors say that they determine “rounded UVI value”: what does that mean? What kind of approximation is used? The authors should clarify how they consider, as an example, the value of UVI=2.3: is it considered as 2, so that it belongs to “Low” category, or as above 2 so that it is in the higher risk class?

We rounded to the nearest integer using a “round half up” approach. We now also state this explicitly in the methods section (lines 102 ff.). Therefore, UVI values of 2.3 and 2.4 still belong to the “low risk” group, whereas a UVI value of 2.5 is already in the “moderate risk” group. This is a common problem when categorizing continuous data.

There’s been a long discussion in the past about the erythemal action spectrum. Although erythemal data were provided by radiometric stations, it would be correct to add to the references the following paper: “Know Your Standard: Clarifying the CIE Erythema Action Spectrum” Ann R. Webb  Harry Slaper  Peter Koepke  Alois W. Schmalwieser (2010) https://doi.org/10.1111/j.1751-1097.2010.00871.x

We now cite this paper in lines 53, 100 and 287 as ref. 15.

Submission Date

29 April 2019

Date of this review

19 May 2019 16:26:18